# Microbiota-Dependent Effects of IL-22

**DOI:** 10.3390/cells9102205

**Published:** 2020-09-29

**Authors:** Morsal Sabihi, Marius Böttcher, Penelope Pelczar, Samuel Huber

**Affiliations:** I. Department of Medicine, University Medical Center Hamburg-Eppendorf, Martinistr. 52, 20246 Hamburg, Germany; m.sabihi@uke.de (M.S.); ma.boettcher@uke.de (M.B.); p.pelczar@uke.de (P.P.)

**Keywords:** IL-22, IL-22BP, IL-22R1, microbiota, cytokines, inflammation

## Abstract

Cytokines are important contributors to immune responses against microbial and environmental threats and are of particular importance at epithelial barriers. These interfaces are continuously exposed to external factors and thus require immune components to both protect the host from pathogen invasion and to regulate overt inflammation. Recently, substantial efforts have been devoted to understanding how cytokines act on certain cells at barrier sites, and why the dysregulation of immune responses may lead to pathogenesis. In particular, the cytokine IL-22 is involved in preserving an intact epithelium, maintaining a balanced microbiota and a functioning defense system against external threats. However, a tight regulation of IL-22 is generally needed, since uncontrolled IL-22 production can lead to the progression of autoimmunity and cancer. Our aim in this review is to summarize novel findings on IL-22 and its interactions with specific microbial stimuli, and subsequently, to understand their contributions to the function of IL-22 and the clinical outcome. We particularly focus on understanding the detrimental effects of dysregulated control of IL-22 in certain disease contexts.

## 1. Introduction

A highly diverse microbiome contributes to the development and maturation of a robust immune system, and is an important factor in maintaining homeostasis at barrier sites [1]. Many beneficial physiological functions arise due to the mutualistic relationship between the host and its microbiome. The host provides a nutrient rich space, whereas resident microbes catabolize and facilitate the acquisition of food molecules. Moreover, components derived from the commensal microbiota prevent the colonization and propagation of pathogenic microbes at these interfaces [2]. Considerably, constant food and microbial antigen exposures at epithelial barriers modulating immune components, have a critical role in the development of the host and progression of immune-related diseases [3].

The host microbiota is closely associated to the regulation of cytokine expression patterns. The composition of the microbiota can be altered easily through changes to the diet, antibiotic intake or substance abuse, and can even undergo diurnal oscillation [3,4,5,6]. Specific factors attributed to the microbiota and related by-products have been discovered to induce dysregulated cytokine expression and inflammation that may be left uncontrolled after a threat has subsided. Under such circumstances, these factors may contribute to the development of chronic inflammation, autoimmunity and cancer by enhancing expression of certain cytokines [7]. Interleukin-22 (IL-22) is one such cytokine that has many protective qualities, but requires constant regulation to prevent overproduction in inflammatory settings [8].

IL-22 is involved in many aspects of the immune system, in that it is an active modulator in preserving an intact epithelium [9,10,11,12], maintaining a balanced microbiota [13] and a functioning defense system against external threats [13]. IL-22 belongs to the IL-10 family, a family of cytokines that is grouped based on their structural similarity, common receptor usage, and similar downstream signaling targets [14]. The description of IL-22 was contemporaneously reported by Renauld and Gurney, and was first referred to as the IL-10-related T cell-derived inducible factor (IL-TIF) [15,16]. IL-22 is an α-helical cytokine, comprising of six α-helices and connecting loops. This cytokine binds to a heterodimeric cell surface receptor complex comprising of the IL-10R2 and IL-22R1 subunits. IL-22R1 defines the specificity of IL-22, as it is predominantly expressed on epithelial cells of the gastrointestinal tract, urogenital tract, lung and skin, but is absent on hematopoietic cells within these tissues [17].

Through the membrane bound IL-22R1, IL-22 is able to signal from the immune system to the tissue. IL-22 is predominantly produced by T helper 17 (Th17), T helper 22 (Th22), and innate lymphoid cells type 3 (ILC3) [18,19,20,21,22,23]. Other cellular sources of IL-22 include γδ T cells, neutrophils and NK T cells [17]. Activation of the IL-22R1 leads to an activation of Janus kinase 1 (Jak1) and non-receptor protein tyrosine kinase 2 (Tyk2) [24]. This activation leads to tyrosine residue phosphorylation of STAT3. In addition to STAT3, STAT1 and STAT5 have also been reported to be activated by IL-22 [24]. Distinct additional pathways have been described to be involved in IL-22 signaling, including Mitogen Activation Protein Kinase (MAPK), Akt [25] and p38 pathways [26].

IL-22 is known to play a role in multiple diseases such as inflammatory bowel disease (IBD), psoriasis, arthritis and cancer [8,17]. In general, IL-22 is considered to have protective effects at barrier sites exposed to external stimuli in an acute setting, however, chronic inflammation can result in dysregulation of IL-22 signaling, promoting overt tissue damage and cancer [27]. Interestingly, the impact of IL-22 in these diseases can be variable, which may be attributed to the microbial factors that affect the actions of this cytokine. In order to dampen possible pathogenic effects, it is crucial to modulate IL-22 by either controlling the production of IL-22 or by blocking it directly via its soluble endogenous receptor, the IL-22 binding protein (IL-22BP) (Figure 1) [27,28].

Originally, IL-22BP was discovered to bind IL-22 and to neutralize its activity in vitro [28]. Later on, it was confirmed that IL-22BP also blocks the activity of IL-22 in vivo in different mouse models [10,27,29]. IL-22BP inhibits IL-22 signaling by limiting the bioavailability of IL-22 to the membrane bound IL-22R1. The binding affinity of IL-22 to IL-22BP is 20- to 1000-fold higher compared to its membrane bound counterpart [28,30,31,32,33,34]. Expression of IL-22BP has been observed in several tissues at various levels, where subsets of CD11b+ conventional dendritic cells (cDCs) are a major source of IL-22BP [27]. IL-22BP expressed by CD11b+ cDCs is considered to be protective as it blocks excessive IL-22 production in chronic inflammatory settings [27], and has been shown to promote antigen sampling through bacterial uptake into Peyer’s patches by inhibiting IL-22 signaling in the follicle-associated epithelium [35]. Eosinophils [36] and CD4+ T cells have also been shown to express IL-22BP in IBD patients, and have been associated as a pathogenic source of the endogenous inhibitor, in that the protective effects of IL-22 are blocked [10]. Studies on the expression patterns of IL-22 and IL-22BP have revealed an inverse relationship during steady state and disease contexts. Specifically, IL-22 expression is most often described to be upregulated during inflammation and as a result of tissue damage. Contrarily, IL-22BP is highly expressed in steady state and downregulated locally upon tissue damage of the epithelial barrier, but is induced once again during recovery [27,34].

In the following sections of this review, we will discuss the role of IL-22 in different organs, with a particular emphasis on signaling and downstream targets. Furthermore, we will focus on the impact of the IL-22/IL-22BP axis, and the different actions of IL-22 with regards to its cellular source and its local environment. Since many reports have discussed the influence of certain microbiota-associated stimuli on the actions of IL-22 signaling, we aim to interpret what is known so far about IL-22 in association to specific microbial compositions and their associated by-products. As the contributions of IL-22 are particularly important at epithelial barriers, we will be concentrating on understanding interactions at interfaces most exposed to external factors, namely the gut, liver, skin, lung, and female genital tract (Figure 2). The search tactic that we utilized during our extensive research in the topic was to look for relevant publications focused on the effect of IL-22 in association with the microbiome in the specific organs mentioned previously.

## 2. IL-22 in the Gastrointestinal Tract

The crosstalk between the epithelium and the local immune cells represents one of the basic features of intestinal homeostasis [37]. Since IL-22 is produced by both innate and adaptive immune cells and specifically targets epithelial cells, it provides a circular link between immunity and mucosal homeostasis. Despite the comparable levels of IL-22 in the intestine of germ free and specific pathogen free mice [38], studies obtained in the last few years have indicated that specific microbial species are capable of promoting or suppressing IL-22. On the one hand, Castleman and colleagues showed that both Gram-positive and -negative commensal bacteria induced the expression of IL-22 by ILC3s. This IL-22 production by ILC3s was not mediated through direct recognition of bacteria, but rather mediated indirectly by myeloid dendritic cells (mDCs) and is partly dependent on IL-23 and IL-1β [39]. On the other hand, another report showed that *Bacillus anthracis* can downregulate IL-22 in ILC3s via an IL-23 mediated MAPK signaling pathway [40].

Similar to the mechanisms inducing IL-22 production in ILC3s, the in vitro studies of Hrdy et al. have shown that exposing a co-culture of dendritic and T cells to probiotic species of the *Lactobacillus* results in the induction of IL-17A and IL-22, while only having a minor impact on regulatory T cells [41]. Conversely, IL-22 signaling pathways can also influence the microbial diversity of the host. While there has been no significant difference in the diversity and community structure of the baseline microbiota between wild-type and *Il22ra1*^−/−^ mice [42], Zenewicz et al. showed that a deficiency of IL-22 in mice alters the commensal microbiota and renders the host more susceptible to colitis. Here, it was shown that IL-22-deficient mice possess a decreased abundance of *Lactobacillus* and an increased abundance of more pathogenic bacteria, such as *Helicobacter* species. Interestingly, the dysbiotic flora of IL-22-deficient mice is transmissible to co-housed wild-type mice and can promote their susceptibility to developing colitis [13]. This study was supported by others, wherein the inhibition of IL-22 signaling also resulted in alterations in the gut microbiota, which subsequently led to an increased serum concentration of metabolites and bacterial products, causing pathology in distant organs [43].

The dysbiosis in IL-22-deficient mice emphasizes the role of IL-22 in the maintenance of healthy gut microbiota compositions. IL-22 has been shown to modulate microbiota compositions indirectly by promoting the growth of commensals. For example, Pham et al. showed that IL-22 signaling promotes epithelial fucosylation, which is important in promoting the growth of commensals and preventing over-colonization by opportunistic pathogens such as *Enterococcus faecalis* [42]. In another study, IL-22 is shown to inhibit the growth of *Clostridium difficile* by enhancing the growth of succinate-consuming bacteria. Here, IL-22 signaling modulates glycosylation of host-derived glycans, which in turn encourages the growth of the commensal bacteria *Phascolarctobacterium spp.* that outcompete *C. difficile* for nutrients [44]. IL-22 also plays a protective role in controlling fungal infections such as Candidiasis. IL-22 was shown to mediate anti-candidal resistance at mucosal surfaces, an activity to which the local microbiota crucially contributes. Zelante et al. described a metabolic pathway whereby tryptophan metabolites from the microbiota balance mucosal reactivity in mice. Switching from sugar to tryptophan as an energy source, the highly adaptive *Lactobacilli* are expanded and produce an aryl hydrocarbon receptor (AhR) ligand—indole-3-aldehyde—that contributes to AhR-dependent IL-22 transcription. The resulting IL-22-dependent balanced mucosal response allows for survival of commensal communities yet provides colonization resistance to the fungi [45,46].

IL-22 does not only impact the commensal microbiota and pathobionts, but also pathogenic bacteria. Indeed, studies have shown that the induction of antimicrobial peptides, which act as an interface between epithelial cells and microbiota, are a critical aspect of IL-22 action against pathogenic infections. IL-22 was found to be very important in early host defenses against *Citrobacter rodentium*. Here, IL-23 is needed to induce IL-22 during the infection, and Reg family proteins induced by IL-22 are pivotal in the survival of the host against the infection [47]. IL-22 mediated upregulation of antimicrobial peptides has also been found to be critical in inducing protection against many other pathogens, such as *Salmonella typhimurium, Klebsiella pneumoniae,* and *Helicobacter pylori* [48,49,50]. In fact, IL-22 stimulates the expression of mucin-1, -3, -10 and-13, that prevents the physical penetration of bacteria and colon epithelial cells in a STAT3-dependent manner [51]. Furthermore, IL-22 confers protection via the STAT3 signaling pathway by inducing the production of other antimicrobial peptides such as BD-2, BD-3, S100A7-9, lipocalin-2, and Reg3β/γ [47,52,53,54]. In particular, the Reg3 protein exerts bactericidal activity against Gram-negative bacteria by interacting with peptidoglycan and is critical for mucosal protection as well as bacterial translocation [55,56,57]. The dysbiosis observed in IL-22-deficient mice was shown to be associated with altered Reg3β and Reg3γ expression reinforcing the concept that IL-22 is important in maintaining the intestinal microbiota via antimicrobial peptide production [13].

Although IL-22 has been shown to be key in regulating barrier function against intestinal bacteria and other insults by upregulating an antimicrobial response, IL-22 induction can be exploited by pathogens such as *Salmonella*. IL-22 boosts its colonization in the inflamed intestine by suppressing commensal *Enterobacteriacae*, which are susceptible to the antimicrobial proteins [58]. A similar role was observed in parasitic infections, particularly in *Toxoplasma*
*gondii* intestinal inflammation induced by oral infection with the ME49 strain [59]. However, the pathogenicity depends on the strain of *T**. gondii* used, the route of infection and the dose administered [60]. Taken together, these findings suggest that the effects of IL-22 on pathogen colonization resistance are highly complex and vary depending on the inflammatory stimuli and the indigenous microbiota.

## 3. IL-22 in the Liver

There is much evidence to suggest that aberrations in the intestinal microbiota can influence extra-colonic sites, a considerable example being the liver. Although the mechanisms regarding how microbiota components affect liver diseases are relatively unknown, a strong correlation has been described between aggravated liver pathology and disturbances to microbiota compositions [61]. This could be attributed to the gut-liver axis, as aberrations in gut homeostasis may result in increased intestinal permeability, systemic translocation of pathogens to liver tissues and subsequent inflammation [62]. Alterations in the gut microbial composition caused by high caloric intake [63] and alcohol consumption [64] are often related to inflammation in the liver as well [65].

Studies have shown that IL-22 is a critical component in modulating homeostasis in the liver. Hepatocytes are the main cellular targets in this context [15,17]. Here, IL-22 is responsible for promoting production of anti-apoptotic, mitogenic and antioxidant molecules in damaged hepatocytes [15,66,67,68]. IL-22 particularly plays a role in preventing steatohepatitis and is crucial for the process of liver regeneration [66,69,70,71].

Like in the gut, there are indications that IL-22 is generally protective in acute settings, where it promotes tissue repair and protection against cellular damage [68]. For example, IL-22 exerts protective effects against *Schistosoma* and *Plasmodia* infections [60,72,73]. Particularly, the IL-22/IL-22BP axis has great implications in *Schistosoma* infections. Here, IL-22 is protective, whereas high levels of IL-22BP are associated with aggravated liver fibrosis and cirrhosis [73].

A growing number of studies indicate that using a fecal microbiota transplantation (FMT) approach with the intention of altering the host’s dysbiotic flora may have therapeutic potential in certain liver-related diseases [63,74,75]. A recent study by Zhou et al. showed that FMT interventions in mice fed with a high-fat diet alleviated steatohepatitis and reduced pro-inflammatory cytokine levels within the liver. This high-fat diet mouse model induced symptoms concurrent with non-alcoholic steatohepatitis (NASH) patients and was used to show that FMT resulted in an increased abundance of beneficial bacteria in the host, namely *Christensenellaceae* and *Lactobacillus*. Subsequently, the FMT recipient mice displayed an increase in butyrate production within the cecal content, a significant downregulation of pro-inflammatory cytokines and a significant upregulation of anti-inflammatory cytokines. Interestingly, IL-22 demonstrated anti-inflammatory properties and appeared to be protective in this setting, reiterating that IL-22 is capable of inhibiting tissue damage and steatosis within the liver [63].

Dysbiosis of the intestinal microbiota and IL-22 have both been implicated in alcohol-related liver disease (ALD) [64,65,76]. Ethanol-exposure causes changes in microbiota diversity and greatly impacts the intestinal epithelial barrier by inducing damage to the mucosal layers and tight junctions, resulting in increased gut leakiness. Consequently, bacterial translocation to the liver increases, inducing inflammation and steatosis [77,78]. Recently, Seo et al. have provided mechanistic insight into how components of the intestinal microbiota can influence the outcome of ALD and the effects of IL-22 in this setting. In stool samples of ALD patients with liver fibrosis and high serum levels of the liver enzymes ALT and AST, the authors found that alcohol consumption resulted in a significant depletion of *Roseburia* species. Daily supplementation of ethanol-fed mice with *Roseburia intestinalis* isolated from patient stool resulted in reduced inflammation and restoration of epithelial barrier integrity, ultimately leading to ameliorated liver disease [64]. Although butyrate production has been described as the contributing factor to alleviated ALD-associated pathogenesis [79], it appears that the protective effects of *Roseburia spp*. are not attributed to this particular function. Specifically, it was found that the flagella on the surface of *R. intestinalis* are effector molecules that upregulate *Occludin* and *Muc2*, resulting in restoration of the gut epithelial barrier in ethanol-fed mice. Moreover, the flagella from these bacteria activate Toll-like receptor 5 (TLR5) signaling and induce an upregulation of IL-22 expression, which in turn induces the production of antimicrobial peptides such as Reg3γ [64]. Consistent with these findings, another report by Hendrikx et al. also credits IL-22 as a major factor in ethanol-induced steatohepatitis, providing evidence that lower levels of microbiota-derived AhR ligands contribute to downregulated IL-22 and Reg3γ production. Here, the authors went further in restoring ethanol-induced downregulation of IL-22 by feeding mice with genetically engineered IL-22-producing *Lactobaccilus reuteri*. Consequently, Reg3γ levels were increased and resulted in diminished liver pathology within these mice [80]. Together, these findings provide strong evidence that IL-22 is indeed implicated in ALD, and that new potential bacterial-based treatments administering IL-22 may be beneficial for the host.

Although many protective effects of IL-22 associated with regulation of microbiota-induced perturbations have been described, there are also studies demonstrating a divergent and pathogenic role of IL-22 in the liver. Particularly, chronic inflammation, caused for example by viral infections, results in dysregulated IL-22 production [81,82]. For instance, IL-22 is increased in patients infected with hepatitis B and C viruses (HBV and HCV), and is correlated with the grade of inflammation in the liver and proliferation of hepatocytes [81,83,84,85]. In this setting, IL-22 has minimal direct antiviral effects, however, excessive IL-22 production results in detrimental effects associated with aberrant inflammation and immune cell-mediated damage, ultimately contributing to liver pathogenesis [81,86]. The expression of IL-22 is also upregulated in acute liver damage caused by ischemia and chemically induced liver damage via administration of acetaminophen. In such settings, where overt IL-22 production contributes to liver injury, a tight regulation via IL-22BP is especially important [29]. In conclusion, evidence indicates that modulation of IL-22 signaling via IL-22BP and the restoration of important microbial components may be helpful in ameliorating inflammation in the liver.

## 4. IL-22 in the Skin

Like most barrier sites, homeostasis and barrier integrity in the skin is regulated by microbial components and immunological factors. The predominant genera making up healthy skin microbiota include *Corynebacterium*, *Propionibacterium*, *Staphylococcus, Streptococcus,* and *Pseudomonas* [87,88]. However, a dysbiosis in the local skin microbiota may cause a dysregulated immune response in the host and contribute to skin pathogenesis [89].

Comorbidities of IBD, obesity and psoriasis indicate a pathogenic link between these diseases that may be attributed to similar immune responses and proinflammatory cytokine profiles [90,91]. Interestingly, in a study by Tan et. al, common intestinal microbiota profiles were identified in these three diseases via 16S rDNA sequencing. The authors provided evidence on the crucial role of gut microbiota on psoriasis, where they found that the low abundance levels of the species *Akkermansia muciniphila*, which is also known to play a role in the pathogenesis of IBD, is significantly reduced in patients with psoriasis [92]. Moreover, it was shown here that, as well as the local skin microbiota, the less apparent gut microbiota also plays a role in inducing immune-mediated pathogenesis.

IL-22 has been demonstrated to play a role in infection and immune-mediated diseases related to the skin. Of note, upregulation or dysregulation of IL-22 results in mostly detrimental effects in the skin, as aberrant control over its production results in hyperplasia of keratinocytes, resulting in conditions such as psoriasis and atopic dermatitis [93]. IL-22 is upregulated in patients with psoriatic lesions and previous studies in imiquimod-induced psoriasis mouse models have reported reduced skin lesions in IL-22-deficient mice, indicating the importance of this cytokine in immune-related skin diseases. Mechanistically, local γδ T cell are one of the predominant producers of IL-22 in the skin and are thus implicated in the pathogenesis of psoriasis [94].

Specifically, many studies on psoriasis have indicated that particular species usually found in healthy skin microbiota, may become opportunistic and contribute to pathogenesis in the skin [95]. When considering the impact of IL-22 within the skin microbiota, the most studied infections are *Staphylococcus* and *Streptococcus*-related infections. Specifically, in *S. aureus* infections affecting atopic dermatitis patients, secreted enterotoxins can inhibit the production of IL-22 from CD4+ T cells. Interestingly, these enterotoxins did not have the same inhibitory effect on CD8+ T cells that produce IL-22. Here, the CD8+ T cells upregulated the secretion of IL-22, indicating that this cell type is less susceptible to *S. aureus* enterotoxins and is responsible for skin pathogenesis in atopic dermatitis patients [96]. Most studies in skin diseases implicate IL-22 as a pathogenic component in the immune response against infections. Interestingly, there are a few exceptions where IL-22 production is shown to be beneficial to the host in this context. For instance, γδ T cell-derived IL-22 has an important role in *S. aureus* infection as it is a critical factor in sufficient bacterial clearance. In a mouse model of skin injury, IL-22-deficient mice had a higher bacterial burden, reduced neutrophil infiltration and production of antimicrobial peptides compared to their wild type counterparts [97]. These studies exemplify how distinct cellular sources of IL-22 contribute differently to the resolution of the same pathogenic source, by either aiding in bacterial clearance or promoting tissue damage.

A few recent reports have also considered the effect of IL-22BP in the skin. IL-22BP is significantly downregulated in skin biopsies taken from psoriasis patients in comparison to healthy controls [98]. Conforming to what is understood about the workings of the IL-22/IL-22BP axis, it has been further demonstrated that IL-22BP acts in a protective manner in skin-related conditions, as reported in mouse models of psoriasis and atopic dermatitis [98,99,100]. Nevertheless, the mechanisms regulating the production of IL-22BP from different cellular sources and, in particular, in response to skin infections is yet to be elucidated.

## 5. IL-22 in the Respiratory Tract

The microbiota in the lung is highly sensitive to intrinsic and environmental factors. Particularly, exposure to allergens and pathogenic microbes have been found to induce a state of dysbiosis in the lung and cause pathogenesis [7,101]. Studies in germ free mice have provided evidence that the local lung microbiota is important in preventing exacerbated pulmonary infections caused by *Klebsiella pneumoniae*, *Streptococcus pneumoniae,* and *Pseudomonas aeruginosa* infections [101].

Like other barrier sites in the body, the expression of IL-22 in the lung has been found to be critical in regulating epithelial repair responses in the lungs after lung injury, conferring a protective role in this organ [102]. Predominant cellular sources of IL-22 in response to infection in the lung include Th17 helper cells, γδ T cells, NK cells, and ILCs [103]. What we know so far about the effects of IL-22 in the lung infections have stemmed from investigations carried out in mouse models of common pathogens responsible for pneumonia development. Particular interest has been shown in the impact of influenza [102,104,105] and super-infection with either *S. aureus* or *S. pneumoniae* on the effects of IL-22 [106,107]. For example, Ivanaov et al. demonstrated that the absence of IL-22 exacerbated lung injury and resulted in diminished epithelial integrity in sub-lethal influenza infection [104]. More recently, it was described that IL-22-mediated tight junction formation also plays a role in conferring protection against influenza infection and bacterial superinfection. However, more emphasis was put on the pathogenic role of IL-22BP in this study, as it blocks the protective effects of IL-22 in these particular disease models [106,108]. Other studies on *K. pneumoniae*, *P. aeruginosa,* and *A. fumigatus* infections have also shown the contributions of IL-22 in defending the lung epithelial barrier [109,110]. As the effects of the IL-22 and IL-22BP axis in the lung have recently been reviewed extensively by Ahn and Prince, we will not go into detail on the microbiota-dependent effects of IL-22 in this organ [103].

## 6. IL-22 in the Female Genital Tract

The epithelial barrier of the female genital tract and its associated commensal microbiota function to protect the host from exposure to an array of external factors, including spermatozoa and pathogenic microbes. The mucosal surface is predominantly populated by *Lactobacillus*, *Prevotella* and *Fusobacteria spp.* [111]. Compared to other organs, research on the effect of IL-22 at epithelial barriers of the genital tract is fairly limited. Although little is known about the IL-22-dependent mechanisms implicated in the female genital tract regarding maintenance of ovarian and uterine function, it is assumed that IL-22 contributes significantly to female reproductive health and has a generally protective nature within this set of organs. Furthermore, IL-22 has been reported to play a role in combating sexually transmitted infections (STIs), intrauterine infections and also contributes to factors involved in pregnancy maintenance [112,113,114].

STIs can be contracted at the mucosal epithelial barrier and can cause severe long-term damage to the epithelial layer [115]. As IL-22 is important for maintenance of epithelial integrity at mucosal barriers, it has been suggested that it may serve in protecting the host from contracting STIs and repressing pathogenesis caused by the microbial threat [112]. Among the diverse group of STIs, infections with *Chlamydia trachomatis* are especially correlated to clinical complications including infertility and increased risk of acquiring other STIs, for example human immunodeficiency virus (HIV) [116]. Significantly, cervical epithelial cells of patients infected with *C. trachomatis* showed elevated IL-17 and IL-22 production in both mouse and human studies [112,117]. Further studies in patients have shown that other STIs, including *Trichomonas vaginalis* and *Neisseria gonorrhea* infections, also induced elevated levels of IL-22 measured in the genital mucosal fluid [112]. Furthermore, Zhao et al. have shown that a combination of IL-22 and TNFα administration induced the production of immunomodulatory cytokines and antimicrobial peptides, resulting in increased epithelial cell survival and inhibition of *C. trachomatis* growth in vitro [118]. However, a handful of studies have reported that although IL-22 is upregulated in the genital tract of infected patients, it appears to have a redundant role in controlling the pathogenesis of *Chlamydia muridarum*, *Candida alibcans,* and *Neisseria gonorrhea* infections in vivo [119,120,121].

Various immune cells produce IL-22 in decidual tissues, making up the mucosal lining of the uterus in preparation of an anticipated pregnancy. Here, IL-22 has an active role in maintaining pregnancy, making it a clinically relevant cytokine during gestational infections. ILC3s have been reported to constitutively produce IL-17 and IL-22 at low frequencies during homeostasis and during pregnancy [122]. Uterine NK cells (uNK) also make up a large proportion of immune cells in decidual tissues during these two stages [118]. Interestingly, uNK cells have been found to produce IL-22 in the decidual tissues, and production is significantly upregulated in tissues of pregnant mice after local challenge with lipopolysaccharide (LPS), an unspecific endotoxin that stimulates immune reactions [113,114]. Another report by Gibbs et al. has provided evidence that mucosal associated invariant T cells (MAIT) cells are present in cervical tissue, and are also responsible for upregulated IL-17 and IL-22 production in response to *Escherichia coli* [123].

Infection-associated inflammation is one of the major risks associated with preterm birth [124,125,126]. Dambaeva et al. showed that intrauterine infections in the late gestational stages of pregnancy promote the production of IL-22 by uNK cells, which have an important protective role in maintaining pregnancy. Experiments involving intrauterine LPS injection resulted in preterm labor and pregnancy loss of all feti in IL-22-deficient mice. In these mice, administration of recombinant IL-22 prevented preterm birth and reduced apoptosis of placental cells [113]. Clinical studies support these findings, as patients with less IL-22 expression in uNK cells in the decidua were found to have higher rates of pregnancy loss [127,128]. This implies that IL-22 could be used as a treatment to prevent preterm birth and combat intrauterine infections [113,129].

Lastly, a recent report by Qi et al. investigated the effects of specific gut microbiota components and metabolic products in patients with polycystic ovary syndrome (PCOS). PCOS is defined by ovarian dysfunction and insulin resistance, and in this study, the authors were able to divulge that reduced IL-22 levels had a major role in the pathogenesis of this disease. Stool samples from patients with PCOS revealed a significantly higher abundance of the bacterial species *Bacteriodes vulgaris* and reduced levels of bile acids compared to matched healthy controls, resulting in aberrant metabolism of acids in these patients. Fecal transplantation of PCOS stool into mouse recipients and monocolonization of mice with *B. vulgaris* resulted in ovarian dysfunction and insulin resistance, mimicking key features associated with PCOS. Particularly, IL-22 production from ILC3s was significantly downregulated in patients with PCOS and in mice treated with *B. vulgaris*, and PCOS-associated symptoms could be reversed via the administration of IL-22 or certain bile acids, such as glycodeoxycholic acid. Additionally, bile acid administration upregulated IL-22 production from ILC3s in mouse models of PCOS, and conversely, the beneficial effect of administering bile acids was reversed in IL-22R1-deficient mice [130]. A follow up report by the same authors then went on further to describe the mechanism as to how IL-22 administration could reverse the pathogenic phenotype in PCOS mouse models. They reported that IL-22 was capable of upregulating the browning of adipose tissue, resulting in regulated insulin sensitivity and ovarian function in a PCOS model [131]. These studies indicate a potentially therapeutic effect of IL-22 in treating PCOS patients.

Conclusively, these studies provide evidence of the critical impact IL-22 has in the female genital tract. Enhancing the protective effects of IL-22 in this organ may aid in combating STIs and complications associated with pregnancy.

Within this review, we have discussed what has been discovered about the actions of IL-22 with regards to microbial stimuli at epithelial barriers exposed to external threats. The following table summarizes the relevant IL-22-associated microbial entities and the cellular sources that are related to IL-22 production in the aforementioned organs (Table 1).

## 7. Summary

Taken together, constant interactions of the immune system with commensal bacteria mainly contribute to mucosal homeostasis at epithelial barriers. Likewise, potentially harmful interactions with pathogenic microbes need to be either avoided or regulated in order to avoid systemic infections or overt inflammation. With regards to this balancing act, IL-22 plays an important role in governing these interactions. On the one hand, we have learnt that components of the microbiota can impact the production and action of IL-22 at epithelial barriers. On the other hand, IL-22 can indirectly influence the composition and abundance of the microbiota, highlighting the dependency of these two factors upon each other.

In this review, we have focused on the actions of IL-22 on commensal microbiota and discussed circumstances where dysbiosis induces perturbed immune responses that may lead to detrimental inflammation and autoimmunity. However, we have not gone into detail about the major impact microbiota and IL-22 have on cancer development. While IL-22 has advantageous effects, such as induction of epithelial cell proliferation and survival during tissue damage, uncontrolled IL-22 activity can promote cancer, as demonstrated in patients and mouse models [27,132,133]. We hypothesize that the dual nature of IL-22 may, in part, be attributed to the cellular source of IL-22. For instance, ILC-derived IL-22 seems to protect against genotoxic stress in the colon [133], whereas IL-22 derived from Th17 cells has been shown to play a pathogenic role in colorectal cancer [134]. Of note, co-production of IL-17 appears to have a major role in the resolution of inflammation and overall tissue damage, particularly in chronic inflammatory settings [135]. Likewise, the different cellular sources of IL-22BP induce contrasting outcomes in disease settings. For example, eosinophil and T cell-derived IL-22BP is pathogenic in an inflammatory setting associated with IBD, however dendritic cell-derived IL-22BP protects against colorectal cancer [10,27,36].

Furthermore, another interesting aspect that requires investigation would be the divergent effects of IL-22 signaling in organs that are traditionally considered to be sterile. A specific example of this would be in the pancreas [136], where IL-22 is known to promote repair and renewal of pancreatic acinar cells during injury [137], but may also lead to the development of pancreatic carcinoma when dysregulated [138]. Recent findings have shown an increase of bacteria in pancreatic carcinoma patients [139,140]. A preclinical study indicated that *H. pylori* colonization in pancreatic cancer cells is associated with the activation of molecular pathways controlling pancreatic cancer growth and progression [141]. An effect of IL-22 on the microbiota seems to be possible, but a solid link between the two has not yet been described within this organ.

Lastly, what is also not completely understood is the connection between the effects of IL-22, the microbiota and graft versus host disease (GvHD). Specifically, the microbiota has been implicated in the development of GvHD [142,143,144,145,146,147,148,149]. However, it remains unclear which microbial factors could be linked to GvHD risk in humans, and whether changes in the microbiota cause the disease or result from it [150]. It is known that IL-22 plays a role in GVHD, but whether its role is beneficial or harmful remains unclear. A recent investigation has provided insight about the effect of IL-22 in this disease, reporting that recipient-derived IL-22 reduced mortality and tissue pathology in the liver and GI tract, whereas donor-derived IL-22 increased mortality and target tissue inflammation [151,152,153,154]. In this case, protective recipient-derived IL-22 was produced by tissue-resident ILCs [155]. It seems likely that there might be a connection between IL-22 and the microbiota in this context, however, there is still a lot left to be investigated in order to establish a clear relationship here.

This review has brought together a collection of reports which help us understand the influence of distinct microbial stimuli in inducing varying effects of IL-22. Ample studies have been discussed here to suggest that there may be many benefits to personalizing future clinical treatments with regards to the patient’s microbial composition and IL-22 requirements in order to combat inflammatory disorders. Definitively, further studies are clearly warranted to decipher the role of the microbiota in influencing the spatiotemporal production of IL-22 and IL-22BP, as well as its impact on disease outcome.

## Figures and Tables

**Figure 1 cells-09-02205-f001:**
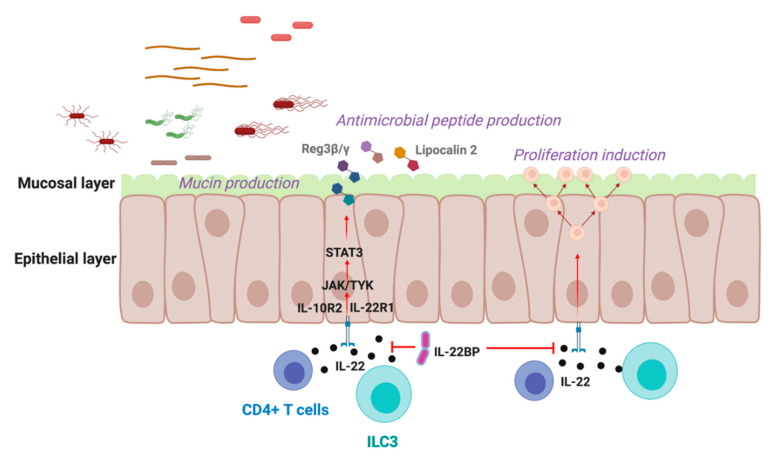
IL-22 regulation at mucosal barriers. IL-22 is a critical cytokine that is highly produced at epithelial barriers. IL-22 is one of many factors that is responsible for fortification of the epithelial layer to prevent the translocation of pathogens and to promote wound repair. IL-22 is predominantly produced by CD4+ T cells and ILC3 populations, among others. Signaling via the heterodimer receptor comprising of IL-10R2 and IL-22R1 initiates JAK and Tyk kinases to activate STAT proteins, which then translocate to the nucleus and induce expression of antimicrobial peptide, mucin, and cell proliferation genes. IL-22BP is the soluble endogenous receptor of IL-22, whose purpose is to inhibit binding of IL-22 to membrane bound IL-22R1. Created with BioRender.com.

**Figure 2 cells-09-02205-f002:**
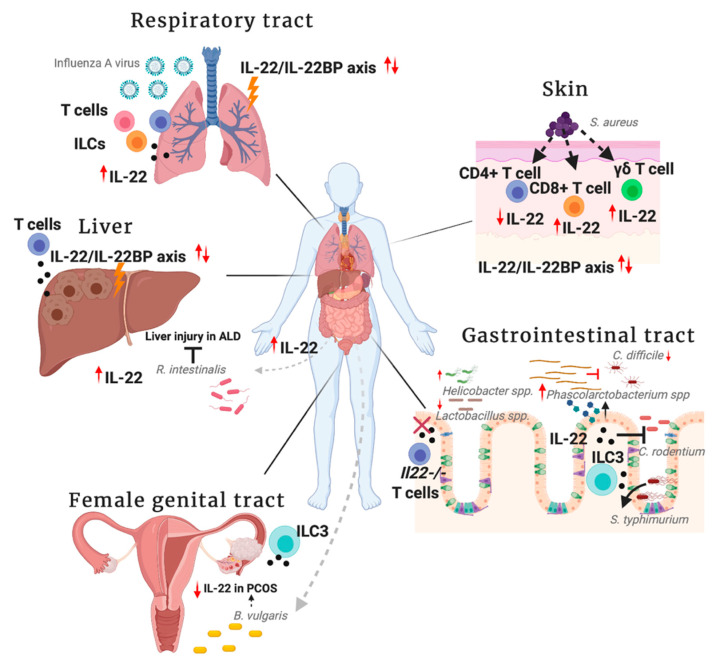
IL-22-associated microbial pathogens. Specific bacterial species, parasites and viruses impact the production of IL-22 at external interfaces and nearby organs. Microbial-derived factors originating from the gastrointestinal tract or other external sites on the host influence the production and subsequent action of the cytokine. Simultaneously, IL-22 in specific microenvironments, derived from different cellular sources also have an impact on colonization and survival of microbial species. The relative expression patterns of IL-22 and IL-22BP may play a role in determining the outcome of certain inflammatory diseases with regards to microbial components that may be implicated. Created with BioRender.com.

**Table 1 cells-09-02205-t001:** Table summarizing microbiota-dependent effects of IL-22 and cellular sources in host organs.

Organ (System)	IL-22-Associated Microbe	Impact of IL-22	Impact of IL-22 on Microbiota	Cellular Source
**Gastrointestinal Tract**	Gram-positive/negative	↑ IL-22 [39]		ILC3
	*Bacillus anthracis*	↓ IL-22 [40]		ILC3
	*Lactobacillus spp.*	↑ IL-22 ↑ IL-17 [41]		CD4+ T cells
		*Il-22*−/− [13]	↑*Helicobacter spp.*	
			↓*Lactobacillus*	
		↑ IL-22 [44]	↓*Clostridium difficile* ↑*Phascolarctobacterium spp.*	
		↑ IL-22 [45]	↓*Candida albicans*	
		↑ IL-22 [47]	↓*Citrobacter rodentium*	
		↑ IL-22 [58] ↑ IL-22 [59]	↑*Salmonella typhimurium* ↑*Toxoplasma gondii*	
**Liver**	*Schistosoma spp.*	↑ IL-22 [73]		
	*Plasmodium spp.*	↑ IL-22 [72]		γδ T cells
	*Christensenellaceae* and *Lactobacillus*	↑ IL-22 in NASH ^1^ [63]		
	*Roseburia intestinalis*	↑ IL-22 in ALD ^2^ [64]		
	*Hepatitis B virus*	↑ IL-22 [83]		CD4+ T cells
	*Hepatitis C virus*	↑ IL-22 [85]		
**Skin**	*Staphylococcus aureus*	↓ IL-22 [96]		CD4+ T cells
		↑ IL-22 [96]		CD8+ T cells
		↑ IL-22 [97]		γδ T cells
**Respiratory tract**	Influenza A virus	↑ IL-22 [104]		αβ T cells, γδ T cells and ILCs
**Female genital tract**	*Chlamydia trachomatis*	↑ IL-22 [112]		
	*Trichomonas vaginalis*	↑ IL-22 [112]		
	*Neisseria gonorrhea*	↑ IL-22 [112]		
	*Escherichia coli*	↑ IL-22 ↑ IL-17 [123]		MAIT cells ^3^
		↑ IL-22 [113]	↓Gram-negative(LPS) ^4^	uNK cells ^5^
	*Bacteriodes vulgaris*	↓ IL-22 in PCOS ^6^ [130]		ILC3

^1^ NASH, non-alcoholic steatohepatitis; ^2^ ALD, alcohol-related liver disease; ^3^ MAIT, mucosal associated invariant T cells; ^4^ LPS, lipopolysaccharide; ^5^ uNK, uterine NK cells; ^6^ PCOS, polycystic ovary syndrome.

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
