# Peer review of "Microbiota-Dependent Effects of IL-22"

_cells, 2020, doi:10.3390/cells9102205_

Round 1

Reviewer 1 Report

Thank you Sabihi et al for the opportunity to review this timely review of microbial-IL22 interactions and their contribution to health and disease. The authors provide a detailed yet focused review describing the biological interactions between various microbial stimuli and IL-22 responses, providing a balanced overview of its regulatory control of mucosal integrity. 

Overall, the manuscript is very well written with a sensible and logical structure. I have a few suggestions mainly to improve clarity for the reader:

  1. There are many topics addressed in this manuscript, which I think is a major advantage, however, I think it would be beneficial for the readers to understand the search strategy used to identify and triage the literature used to craft this manuscript. Was this performed with systematically? 
  2. Figure 1 nicely summarises the context of the review, however it is quite detailed to a point where the main message is lost. I would recommend having a distinct panel for each organ system of interest and being more consistent with the text/font (i.e. size, font, colour). I would also consider removing some of the graphics to improve clarity. Whilst not as visually engaging, often a flow chart can provide more clarity for the reader .
  3. There are some complex biological processes explained throughout the manuscript, the majority of which describe the intracellular signalling processes for IL-22. I wonder is Figure 1 (which basically states the same as Table 1) could be amended to focus on the illustration of IL-22 regulation/production at mucosal interfaces?
  4. It may be important/relevant to include some discussion on the recent findings linking Il-22/ILCs and GvHD given the relevant immune/microbial/mucosal interaction. 
  5. The summary is great, very succinct and covers key points to manuscript. Could it be expanded slightly to discuss how this knowledge could be used to improve/change clinical practice? 

Author Response

Thank you Sabihi et al for the opportunity to review this timely review of microbial-IL22 interactions and their contribution to health and disease. The authors provide a detailed yet focused review describing the biological interactions between various microbial stimuli and IL-22 responses, providing a balanced overview of its regulatory control of mucosal integrity. 

Overall, the manuscript is very well written with a sensible and logical structure. I have a few suggestions mainly to improve clarity for the reader:

  1. There are many topics addressed in this manuscript, which I think is a major advantage, however, I think it would be beneficial for the readers to understand the search strategy used to identify and triage the literature used to craft this manuscript. Was this performed with systematically? 

We thank the reviewer for this question. We performed the literature search systematically and we have indicated what we will be focusing on and why in the manuscript. Additionally, we have provided a clearer explanation as to how we carried out the literature search.

‘In the following sections of this review, we will discuss the role of IL-22 in different organs, with a particular emphasis on signaling and downstream targets. Furthermore, we will focus on the impact of the IL-22/IL-22BP axis, and the different actions of IL-22 with regards to its cellular source and its local environment. Since many reports have discussed the influence of certain microbiota-associated stimuli on the actions of IL-22 signaling, we aim to interpret what is known so far about IL-22 in association to specific microbial compositions and their associated by-products. As the contributions of IL-22 are particularly important at epithelial barriers, we will be concentrating on understanding interactions at interfaces most exposed to external factors, namely the gut, liver, skin, lung and female genital tract. The search tactic that we utilized during our extensive research in the topic was to look for relevant publications focused on the effect of IL-22 in association with the microbiome in the specific organs mentioned previously.’ (Page 4, Line 97-107)

  1. Figure 1 nicely summarises the context of the review, however it is quite detailed to a point where the main message is lost. I would recommend having a distinct panel for each organ system of interest and being more consistent with the text/font (i.e. size, font, colour). I would also consider removing some of the graphics to improve clarity. Whilst not as visually engaging, often a flow chart can provide more clarity for the reader.

  1. There are some complex biological processes explained throughout the manuscript, the majority of which describe the intracellular signalling processes for IL-22. I wonder is Figure 1 (which basically states the same as Table 1) could be amended to focus on the illustration of IL-22 regulation/production at mucosal interfaces?

We agree with this reviewer and have changed figure 1 according to what has been suggested in point 2. Specifically, we have decided to split the figure into two in accordance to what has been suggested in point 3. Figure 1 will focus on IL-22 signaling at mucosal surfaces. Figure 2, which is a modified version of the original Figure 1 has now been edited to make the font and sizes of text consistent and has been edited to reduce the amount of information to only the main points in the review.

  1. It may be important/relevant to include some discussion on the recent findings linking Il-22/ILCs and GvHD given the relevant immune/microbial/mucosal interaction. 

We agree with this reviewer that IL-22/ ILCs also plays a role in GvHD. And we thus expanded the discussion of the manuscript to include this topic.

‘Lastly, what is also not completely understood is the connection between the effects of IL-22, the microbiota and Graft versus Host Disease (GvHD).  Specifically, the microbiota has been implicated in the development of GvHD [143-150]. However, it remains unclear which microbial factors could be linked to GvHD risk in humans, and whether changes in the microbiota cause the disease or result from it [151]. It is known that IL-22 plays a role in GVHD, but whether it’s role is beneficial or harmful remains unclear. A recent investigation has provided insight about the effect of IL-22 in this disease, reporting that recipient-derived IL-22 reduced mortality and tissue pathology in the liver and GI tract, whereas donor-derived IL-22 increased mortality and target tissue inflammation [152-155]. In this case, protective recipient-derived IL-22 was produced by tissue-resident ILCs [156]. It seems likely that there might be a connection between IL-22 and the microbiota in this context, however, there is still a lot left to be investigated in order to establish a clear relationship here.’ (Page 13, Line 440-450)

  1. The summary is great, very succinct and covers key points to manuscript. Could it be expanded slightly to discuss how this knowledge could be used to improve/change clinical practice? 

We thank this reviewer for the very positive assessment of our work. As requested we have expanded the summary slightly to discuss how this knowledge could be used to improve/change clinical practice.

‘This review has brought together a collection of reports which help us understand the influence of distinct microbial stimuli in inducing varying effects of IL-22. Ample studies have been provided to suggest that there may be many benefits to personalizing future clinical treatments with regards to the patients microbial composition and IL-22 requirements to combat inflammatory disorders.’ (Page 13-14, Line 451-459 )

Please refer to the attachment, which is an updated version of the manuscript.

Reviewer 2 Report

In this review, Sabihi et al. compile the main findings on the interaction of IL-22 production with certain stimuli related to the microbiota, in the gastrointestinal tract, liver, respiratory tract, skin and the female genital tract.
The work is well written and has great interest today.
Minor comments
-Authors could include studies of IL-22 in pancreas that exist in the literature. Although it has not been shown that the pancreas has its own microbiota, there is some evidence that bacterial metabolites of the intestine can affect this organ.
-On page 6 line 203, authors should clarify the expression
"leading to depletion of mucosal layers"

Author Response

Reviewer 2

In this review, Sabihi et al. compile the main findings on the interaction of IL-22 production with certain stimuli related to the microbiota, in the gastrointestinal tract, liver, respiratory tract, skin and the female genital tract.
The work is well written and has great interest today.

Minor comments

-Authors could include studies of IL-22 in pancreas that exist in the literature. Although it has not been shown that the pancreas has its own microbiota, there is some evidence that bacterial metabolites of the intestine can affect this organ.

 We thank the reviewer for their comments.

We agree with this reviewer that IL-22 might also play a role in the pancreas. And we thus expanded the discussion of the manuscript to include organs that we didn’t get a chance to discuss in the main text that may be relevant when considering the effects of IL-22 and the microbiota.

‘Furthermore, another interesting aspect that requires investigation would be the divergent effects of IL-22 signaling in organs that are traditionally considered to be sterile. A specific example of this would be in the pancreas [137], where IL22 is known to promote repair and renewal of pancreatic acinar cells during injury [138], but may also lead to the development of pancreatic carcinoma when dysregulated [139]. Recent findings have shown an increase of bacteria in pancreatic carcinoma patients [140,141]. A preclinical study indicated that H. pylori colonization in pancreatic cancer cells is associated with activation of molecular pathways controlling pancreatic cancer growth and progression [142]. An effect of IL-22 on the microbiota seems to be possible, but a solid link between the two has not yet been described within this organ.’ (Page 13, Line 431-439)

-On page 6 line 203, authors should clarify the expression
"leading to depletion of mucosal layers"

We have clarified this sentence. We have changed the text from: ‘Ethanol-exposure causes changes in microbiota diversity and greatly impacts the intestinal epithelial barrier, leading to depletion of mucosal layers and gut leakiness.’ to ‘Ethanol-exposure causes changes in microbiota diversity and greatly impacts the intestinal epithelial barrier by inducing damage to the mucosal layers and tight junctions, resulting in increased gut leakiness.(page 7, line 226-228)

Please refer to attachment, which is an updated version of the manuscript.
